# Adhesion-regulated junction slippage controls cell intercalation dynamics in an Apposed-Cortex Adhesion Model

**Alexander Nestor-Bergmann**[1]*, **Guy B. Blanchard**[1], **Nathan Hervieux**[1], **Alexander G. Fletcher**[2], **Jocelyn Étienne**[3]*, **Bénédicte Sanson**[2]

1 Department of Physiology, Development and Neuroscience, University of Cambridge, Cambridge, United Kingdom, 2 School of Mathematics and Statistics and Bateson Centre, University of Sheffield, Sheffield, United Kingdom, 3 LIPHY, CNRS, Univ. Grenoble Alpes, CNRS, LIPhy, 38000 Grenoble, France

* an529@cam.ac.uk (AN-B); jocelyn.etienne@univ-grenoble-alpes.fr (JE)

## Abstract

Cell intercalation is a key cell behaviour of morphogenesis and wound healing, where local cell neighbour exchanges can cause dramatic tissue deformations such as body axis extension. Substantial experimental work has identified the key molecular players facilitating intercalation, but there remains a lack of consensus and understanding of their physical roles. Existing biophysical models that represent cell-cell contacts with single edges cannot study cell neighbour exchange as a continuous process, where neighbouring cell cortices must uncouple. Here, we develop an Apposed-Cortex Adhesion Model (ACAM) to understand active cell intercalation behaviours in the context of a 2D epithelial tissue. The junctional actomyosin cortex of every cell is modelled as a continuous viscoelastic rope-loop, explicitly representing cortices facing each other at bicellular junctions and the adhesion molecules that couple them. The model parameters relate directly to the properties of the key subcellular players that drive dynamics, providing a multi-scale understanding of cell behaviours. We show that active cell neighbour exchanges can be driven by purely junctional mechanisms. Active contractility and cortical turnover in a single bicellular junction are sufficient to shrink and remove a junction. Next, a new, orthogonal junction extends passively. The ACAM reveals how the turnover of adhesion molecules regulates tension transmission and junction deformation rates by controlling slippage between apposed cell cortices. The model additionally predicts that rosettes, which form when a vertex becomes common to many cells, are more likely to occur in actively intercalating tissues with strong friction from adhesion molecules.

## Author summary

During development tissues undergo dramatic shape changes to build and reshape organs. In many instances, these tissue-level deformations are driven by the active reorganisation of the constituent cells. This intercalation process involves multiple cell neighbour exchanges, where an interface shared between two cells is removed and a new interface is

**Data Availability Statement:** The source code implementing the apposed-cortex model is publicly available for use at https://gricad-gitlab.univ-

grenoble-alpes.fr/etiennej/acam Zenodo record: https://zenodo.org/record/5838249. Documentation and quick-start tutorials can be found at appcom.readthedocs.io.

**Funding:** This work was supported by a Wellcome Trust Investigator Award to BS (099234Z12Z and 207553Z17Z). ANB was also supported by a University of Cambridge Herchel Smith Fund Postdoctoral Fellowship. AGF was supported by a Vice-Chancellor's Fellowship from the University of Sheffield and BBSRC grant BB/R016925/1. All authors were additionally supported by a PICS CNRS travel grant and ANR-11-LABX-0030 'Tec21' grant. The STED microscope was funded by BBSRC grant BB/R000395/1. The Cactus cluster of the CIMENT infrastructure, where the computations were run, was supported by the Rhône-Alpes region (Grant CPER07\_13 CIRA) The funders had no role in study design, data collection and analysis, decision to publish, or preparation of the manuscript.

**Competing interests:** The authors have declared that no competing interests exist.

grown. The key molecular players involved in neighbour exchanges, such as contractile motors proteins and adhesion complexes, are now well-known. However, how their physical properties facilitate the process remains poorly understood. For example, how do cells maintain sufficient adhesive contact while actively uncoupling from one another? Then, how does a new interface grow in a contractile environment? Many existing biophysical models cannot answer such questions, due to representing shared cell interfaces as discrete elements that cannot uncouple. In this paper, we develop a model that represents cell cortices as contractile rope-loops coupled by adhesions. We outline the conditions required for successful neighbour exchanges, in terms of the properties of the known molecules that drive the process. The model predicts that tissue dynamics depend strongly on the ability of neighbouring cortices to slip relative to one another, which is regulated by adhesion turnover.

## Introduction

In both developing and adult animal tissues, cell rearrangements are a common mechanism by which cells actively drive tissue deformation and passively relax stress [1–5]. In epithelia, directed neighbour exchanges between four cells (known as T1 transitions; Fig 1A) are a minimal example of rearrangement that is characterised by the shortening of a shared cell-cell contact, to the point where four cells meet (forming a 4-way vertex), followed by the formation of a new cell-cell contact between previously non-neighbouring cells. This intercalation process can be found throughout development, for example during fish epiboly, mammalian and insect axis extension and hair follicle formation and amphibian and fish neural folding [6–9]. However, surprisingly little is known about the mechanical behaviour of cortical material and adhesions during intercalation. For example, do cell vertices act as physical barriers that keep material constrained within a junction, or can cortical actomyosin be moved between junctions past the tricellular vertex? In the former case, junction length changes can occur only from elastic deformations and eventual actin disassembly [10]. In the latter case, neighbouring junctions can exchange material to commensurately shorten and elongate [11]. Furthermore, it is not clear how the properties of adhesion molecules facilitate uncoupling of connected cell cortices to allow remodelling, while preserving tissue integrity.

Much work has been devoted characterising the localisation of the subcellular constituents involved in driving neighbour exchanges across experimental models. Active mechanisms are known to be involved in many cases of junction shrinkage: contractile Myosin II loads the junctional cortex in invertebrate [12–17] and vertebrate models [18–20], while other mechanisms, such as pools of medial actomyosin [21, 22], appear to be tissue-specific. In many cases, the exact physical roles of subcellular molecules remains unclear, with new proposed roles for junctional Myosin as a ratchet [11] beyond its long-established role in contraction [13]. It is currently also unclear whether such active mechanisms are required to drive the growth of a new junction, or if extension may be energetically favourable and follow passively. Following shrinkage, before extension can occur, the 4-way vertex must be resolved. Delays or failures in resolution can lead to the formation of higher-order vertices, shared by many cells, known as rosettes (Fig 1A) [23, 24]. A higher prevalence of rosette structures has been linked to perturbations in the mechanical properties of tissues and defects in tissue reshaping during morphogenesis [25, 26]. However we have little information about what defines the 4-way resolution timescale and whether it is actively tuned to prevent topological defects in a tissue.

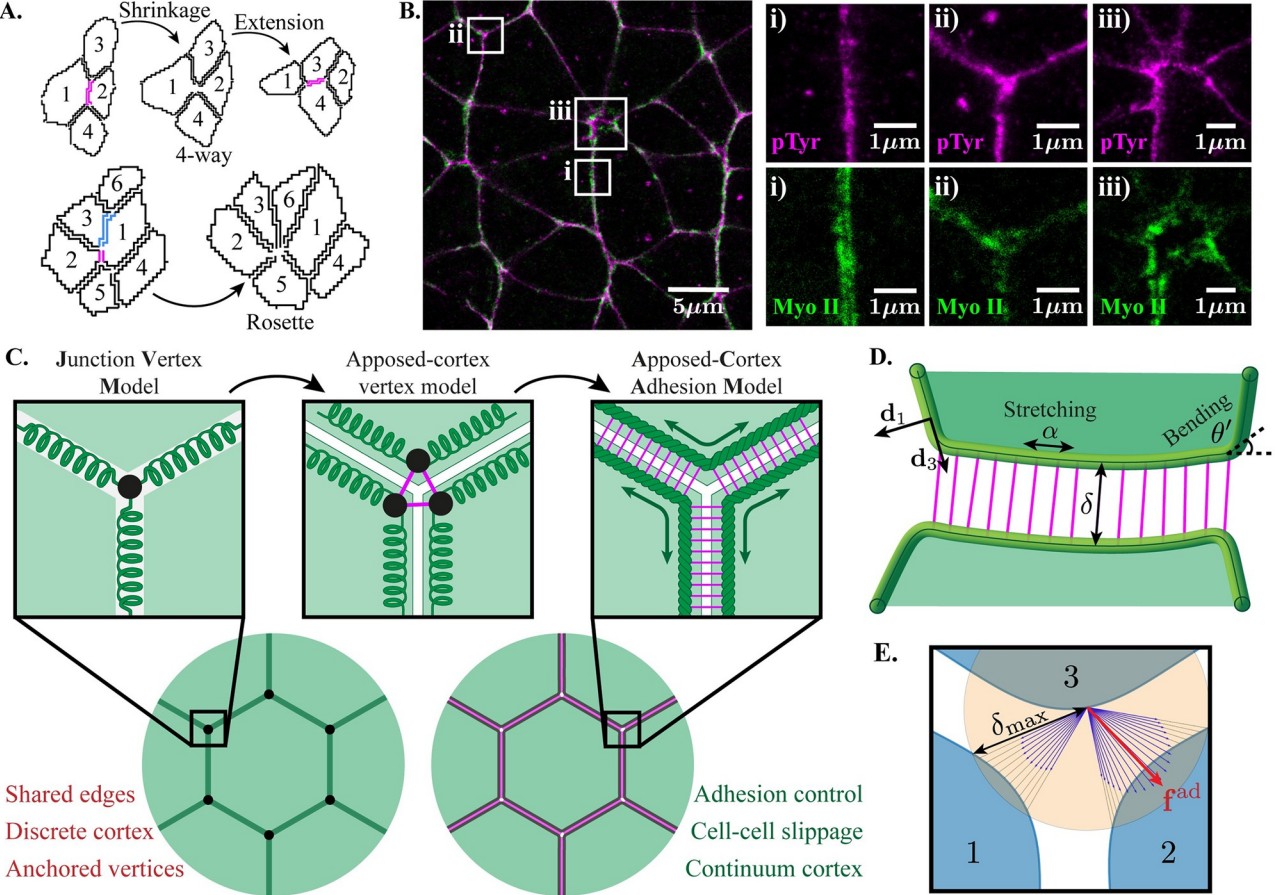

**Fig 1. The geometry and mechanics of apposed cortices. (A)** Example of a T1 transition (top) and a rosette (bottom), segmented from movies of stage 7–8 *Drosophila* embryos [17]. The junctions undergoing shrinkage or elongation are coloured. **(B)** Two-colour STED microscopy image showing the adherens junctions (pTyr; magenta) and cortical Myosin II (green) in intercalating germ-band cells from stage 8 *Drosophila* embryos. The close ups highlight a bicellular junction (i), a vertex (ii), and a rosette centre (iii). **(C)** Extending vertex-based models to allow explicit modelling of neighbour exchanges. Cortices cannot uncouple in vertex models, although Junction Vertex Models (JVMs) [31, 41] allow edges to have individual properties. Distinction of the two cortices within shared edges allows uncoupling, but vertices remain as fixed material points. Replacing vertex anchors with explicit adhesions allows continuous rearrangements, with slippage between neighbouring cells as cortical material can pass between junctions. **(D)** Mathematical representation of a tissue in the Apposed-Cortex Adhesion Model. Cell cortices are represented as continuous rope-loops, with resistance to stretching and bending and viscous turnover on timescales $\tau_{\text{cor}}$. Neighbouring cells are coupled by explicit adhesion spring elements, that turn over with timescale $\tau_{\text{adh}}$. **(E)** A simulated vertex in the Apposed-Cortex Adhesion Model. Discretised cortex nodes without adhesion bonds can connect to neighbouring cortex nodes within $\delta_{\max}$ (e.g. orange shading around a point on cell 3; blue arrows indicate the inverse lengths of possible adhesions). The total adhesion force at this location is $\mathbf{f}_{\text{adh}}$, which is balanced by internal forces in the cortex.

Discrete vertex-based models are perhaps the most popular mechanical models of epithelial tissues [2, 27–31]. These models are simple and computationally inexpensive, yet successfully capture a remarkable variety of tissue behaviour. However, they often suffer the following limitations: (i) phenomenological mechanics, preventing a mechanistic understanding in terms of key biological players; (ii) single edges represent both apposed cortices in a bicellular junction (see [32] as an exception). Neighbouring cells can therefore not uncouple from one another and there is no dissipation associated with their slippage—a crucial aspect of many cell behaviours, such as intercalation and division. In particular, the parameters of vertex models do not have obvious mechanistic relationships to the properties of adhesion molecules. Adhesion dynamics can produce emergent drag forces [33] that regulate the dynamics of cell–substrate interactions by penalising sliding behaviour [34]. By explicitly modelling adhesions molecules,

for example, cell-level phenomenology can be described in terms of the mechanical properties of known elements at the subcellular scale [35].

Since neighbouring cortices cannot uncouple, discrete shared-edge models cannot study cell neighbour exchange as a continuous process. However, the separated cortices of cells in *Nematostella vectensis* have been successfully modelled using a discrete network of multiple springs [36], though this discrete model is limited to the behaviour of springs in series. The immersed boundary method successfully models apposed cortices as continuum structures [37], providing insight to tumour growth and limb bud morphogenesis [38, 39]. However, the model requires the presence of an unphysical incompressible fluid between cell cortices, requiring artificial source and sink terms to control growth [40]. A fluid-free cell boundary model, that is more faithful to existing physical quantities, is likely to be more appropriate for simulating epithelial tissue behaviour.

In this paper, we present an Apposed-Cortex Adhesion Model of an epithelial tissue in terms of parameters that relate directly to the properties of subcellular mechanical constituents. The junctional cortex is modelled as a continuous viscoelastic rope-loop with explicit adhesions between the cortices of different cells. Slippage between apposed cortices and the displacement of vertices along the cortices are emergent features of the model. The model is used to understand the minimal conditions under which cell neighbour exchanges (junction shrinkage, resolution and extension) can be driven by active subcellular contractility in the junctional actin network. We further demonstrate that adhesion dynamics are a key feature in regulating the dynamics of an active tissue.

## Results

We set out to construct from first principles a continuum model of junctional actomyosin and the associated adhesion molecules during cell rearrangement. The model is inspired and validated by *in-vivo* experimental work in the *Drosophila* germ-band—the system in which polarised cell intercalation is best understood. However, we use these comparisons as validation only. We aim to derive general properties of rearrangement as a continuous process, identifying the minimal ingredients and key physical principles that are not bound to specific tissue types.

### Introduction to the apposed-cortex model: Passive mechanics

Across multiple tissue types, actomyosin and adhesion molecules, such as E-cadherin, are key mechanical constituents that modulate tissue dynamics [4, 7]. In order to gain a mechanistic understanding of how their properties regulate neighbour exchanges, a model must have parameters with direct associations to these components. We confirm, using high-resolution microscopy (STED) in the *Drosophila* germ-band, that neighbouring cell cortices sit apposed, with Myosin loading each cortex (notice that the heterogeneity in Myosin II signal is not matched between apposed cortices in Fig 1Bi). Junction vertex models (JVMs), where cell junctions are represented as discrete viscoelastic elements (Fig 1C), have been introduced to model situations where individual junctions have distinct mechanical properties [31, 41]. This is an extension to traditional Cell Vertex Models (CVMs), where junctions share whole-cell properties, such as area and perimeter constraints. However, in Fig 1Bii and 1Biii, we also see that cortices are coupled by adhesion molecules and show variable separation around cell vertices. Intuitively, it is clear that adhesions must uncouple to allow neighbouring cortices to slide and move apart during neighbour exchanges. By modelling bicellular junctions as single edges, anchored by vertices, JVMs do not have direct access to the properties of adhesions, nor can they exhibit the cell-cell shear and cortex-cortex separation required during neighbour

exchange. We therefore model a vertex as the geometric point where three or more cells are coupled by adhesions, rather than a material point (Fig 1C). We term this class of model an "Apposed-Cortex Adhesion Model" (ACAM).

A full description of the model is presented in S1 Text. Here, we summarise how the model is constructed and introduce key parameters that will be shown to regulate intercalation behaviour. The model is composed of simple physical elements, each with a minimal number of parameters which are taken constant and, whenever possible, matched to experimental observations. An adhesion bond is described as an elastic element between two points of neighbouring cortices. Its rest length is the constant $\delta_0$, which we have used to nondimensionalise all lengths. This sets the equilibrium inter-cortical distance between two neighbouring cells at rest. For reference, inter-cortical distances have been measured to be between 30–40 nm (estimated using scale bar in Fig 7 of [42]). The apical cortex of a cell is modelled as a planar viscoelastic thread forming a rope-loop. It has been argued that the observed relaxation time of the cortex is likely to be driven by actin turnover on timescales of $\tau_{cor} \sim 50$ s [10, 43]. Given that this is more than an order of magnitude shorter than the timescale of cell rearrangement ($\sim 15$ min [17]), we assume that the cortex behaves as a viscoelastic fluid in the physiological timescale [44] and nondimensionalise time relative to $\tau_{cor}$. This separation of time scales allows us to simulate viscoelasticity by a complete relaxation of residual elastic stress at each timestep of the simulation (see S1 Text for further details).

**The cell cortex.** We assume that the cortex is the leading mechanical driver of cell behaviour [45] and begin by considering its passive mechanical properties. Rather than representing each cell junction as a discrete viscoelastic element, we model the entire apical cortex as a continuous viscoelastic rope-loop (Fig 1D). Due to its thickness and actin/cross-linker composition, the cortex is assumed to resist bending and extension, with moduli $B$ and $E$ respectively. The instantaneous elastic behaviour of the cortex follows that of an extensible, unshearable and torsion-free rod, with energy [46]

$$\mathscr{U}(\mathscr{C}) = \oint_{\mathscr{C}} \left[ \frac{1}{2} \kappa^2 c(S)^2 + \frac{1}{2} \varepsilon(S)^2 \right] \mathrm{d}S, \tag{1}$$

where $S$ is a curvilinear Lagrangian coordinate parameterising position along the cortex centreline in its reference configuration $\mathscr{C}$. The resistance to extension is captured with respect to local strain $\varepsilon = \alpha - 1$, with stretch $\alpha$. Bending is penalised with reference to the local curvature, $c = \partial\theta/\partial S$, where $\theta$ is the deflection of the cortex (Fig 1D). The relative resistance to bending vs. stretching is encoded by the dimensionless ratio of their moduli, $\kappa = \sqrt{B/E}/\delta_0$, which we will demonstrate regulates equilibrium cell packing geometries. At rest, an isolated segment of cortex would lie as a straight rod.

**Adhesions.** In addition to the passive constitutive material properties of the cortex, we consider forces acting on the cortex due to adhesions. Adhesion is modelled as a single agent that accounts for the composite effect of all molecules associated with the adhesion complex, such as E-cadherin, $\alpha$- and $\beta$-catenin and vinculin [47]. An adhesion bond coupling two cortices together is modelled as a simple Hookean spring (Fig 1D), with energy

$$\mathscr{W}(s, s') = \begin{cases} \frac{1}{2} \omega(\delta(s, s') - 1)^2 & \text{if } \delta(s, s') \leq \delta_{max}, \\ \frac{1}{2} \omega(\delta_{max} - 1)^2 & \text{otherwise}, \end{cases} \tag{2}$$

where $\delta(s, s')$ is the distance function between two points, $s$ and $s'$ of cortices $c$ and $c'$, respectively, in their current configuration (denoted $c$ and parameterised by $s$), $\omega$ is the

dimensionless adhesion strength, equal to the stiffness of a bond scaled by the extensional stiffness of the cortex, $E$. The adhesion energy saturates such that adhesion bonds do not exert force when $\delta > \delta_{max}$. This simpler behaviour compared to Bell's law [48] makes interpretation of results more straightforward. Unbound locations attempt to re-bind to neighbouring cortices lying within distance $\delta_{max}$ according to a probability density function, $\Phi$, described in S1 Text. The dimensionless adhesion energy density at a location $s$ on cortex $c$

$$u_{adh}(s) = \frac{1}{2} \sum_{c',c' \neq c} \oint_{c'} \rho(S(s), S'(s')) \mathcal{W}(s, s') \, ds' \qquad (3)$$

where the summation is over all other cells in the tissue and $\rho$ corresponds to $\Phi$ when adhesion binding and unbinding is instantaneous. In this study, we allow adhesion unbinding to have a finite timescale, $\tau_{adh}$. This timescale is not usually considered in tissue-scale models. It represents the average bond lifetime i.e. how long a bond persists for before it disassociates. We will demonstrate how the balance between cortical and adhesion turnover regulates tissue dynamics. The adhesion density function then obeys a time evolution of the form

$$\tau_{adh} \frac{\partial \rho}{\partial t}(S, S'; t) = \Phi(s(S), s'(S')) - \rho(S, S'; t). \qquad (4)$$

Note that $\rho$ pertains to material points and is thus taken with respect to the reference configuration, $\mathcal{C}$, in contrast to $\Phi$ which depends on the current configuration, $c$. For simplicity, in this introductory paper, we do not consider $\tau_{adh}$ to be affected by local stress because the response is likely to depend on the specific tissue context [49–51] and we look for a general and minimal set of parameters that regulate tissue dynamics. However, stress-dependent dynamics are relatively simple to explicitly encode in this model.

**Local force balance along the cortex.** For a tissue of $N$ cells, the total dimensionless energy is given by

$$\mathcal{U}_{tot} = \sum_{0 < i \leq N} \left( \mathcal{U}_{cor}(\mathcal{C}_i) + \oint_{c_i} u_{adh}(s_i) \, ds_i \right) \qquad (5)$$

Along the cortex of a cell, the force from adhesion molecules is balanced locally by internal forces from cortex bending and stretching, satisfying the balance of linear momentum in the reference configuration:

$$\frac{\partial \mathbf{n}(S)}{\partial S} + \mathbf{f}_{adh}(S) = \mathbf{0}, \qquad (6)$$

where $\mathbf{f}_{adh}(S)$ is the total force from all adhesions connected to cortex location $S$. The internal cortical force, $\mathbf{n}$, follows from Eq (6) (see S1 Text)

$$\mathbf{n} = -\frac{\kappa^2}{\alpha} \frac{\partial c}{\partial S} \mathbf{d}_1 + (\alpha - 1)\mathbf{d}_3, \qquad (7)$$

where $(\mathbf{d}_1, \mathbf{d}_3)$ are orthonormal vectors in the normal and tangential directions along the cortex (Fig 1D).

The numerical scheme requires a discretisation of the cortex continuum. Adhesions then function as discrete elements connecting discretisation nodes between cortices (Fig 1E). New bonds are formed instantaneously, since measurements indicate adhesion recovery times are short, $\sim 20$ s [52, 53], and persist for an average time $\tau_{adh}$. Note that a node may have connections to multiple nodes on other cortices, but new connections are sought only when the node is unpaired.

We generate tissues with multiple cells, coupled by adhesions, each satisfying the local balance of cortical material and adhesion forces, Eq (6). The parameters selected for the simulations that follow are given in S1 Table.

## Vertex geometry is determined by the cortex bending stiffness relative to adhesion strength

The constitutive properties of the cortex are determined by its bending, $B$, and extensional, $E$, moduli, which we capture by their dimensionless ratio, $\kappa^2$. This ratio relates to the length of the curved opening around cell vertices, which can be used to parameterise the model. Consider a single cell enclosed by a hexagonal boundary to which it adheres (Fig 2A). The size, $\delta^{\text{vert}}$, of the opening at the vertices of the hexagonal boundary is determined by $\kappa$, relative to the adhesion strength, $\omega$: increasing $\kappa$ increases the penalty for having sharp corners (large curvature), requiring stronger adhesion, $\omega$, to close the vertex. The size of the opening at vertices is therefore a geometric feature that characterises the passive mechanical properties of a tissue in this model. Mapping this geometric feature across $(\kappa, \omega)$ parameter space we identify isolines of constant $\kappa^2/\omega$, with equal $\delta^{\text{vert}}$ (Fig 2B). The cortex extensional modulus, $E$, cancels out in the ratio $\kappa^2/\omega$ (see Eq (24) of S1 Text), such that the geometry around vertices is prescribed by the ratio of cortex bending modulus to adhesion modulus.

To our knowledge, there are no explicit measurements of the size of vertex openings in the literature. Our STED imaging of adherens junction in the germ-band indicates that vertex sizes are comparable to the bicellular spacing (Fig 1B, panel ii, pTyr). Higher resolution imaging with electron microscopy of other *Drosophila* tissues also suggests that vertices are tightly closed (see e.g. Fig 7 of [42]). We therefore choose order-of-magnitude estimates $\kappa = 0.01$, $\omega = 0.05$, which give $\delta^{\text{vert}} \sim 1.43$ (relative to the bicellular spacing; Fig 2B, red marker). S1(A) Fig shows some representative vertex structures across $\kappa^2/\omega$ samples.

## Active mechanics in the apposed-cortex model

In addition to having passive resistance to deformation, cells are active materials that can generate stresses within the cortex. We use the theory of morphoelasticity [46] to incorporate this behaviour. In passive materials, the shape of an object is determined by its resistance to the external forces and boundary conditions that deform it from a reference (stress-free) configuration. For example, the stiffness of a spring determines the stretch, $\alpha$, produced by a pulling force. In a morphoelastic (active) material, deformations can be generated by the material itself. In the spring example, we can imagine that the spring exerts an active contraction, $\gamma$, reducing its resting length (Fig 2C). Now, in the presence of the same external force, the active spring does not extend as far as the passive spring, since there is resistance from both the stiffness and the pre-stress from the active contraction. We employ this theory in the mechanical description of the cortex (Fig 2D; see S1 Text for a detailed theoretical description). Active stresses within the cortex deform it from its undeformed shape (parameterised by $\hat{S}$) to a new reference shape (parameterised by $S$), locally stretching/compressing by a factor:

$$\gamma = \frac{\partial S}{\partial \hat{S}}, \tag{8}$$

representing local active length changes. Since molecular motors tend to exert contractile forces, we deal only with the case of active contraction, $\gamma < 1$. This generates a positive pre-stress that represents loading regions of the cortex with active Myosin II, the main driver of contractility in many tissues. The effect of loading contractile machinery on the cortex can hence be conceptualised as substituting cortex segments with segments that have smaller rest

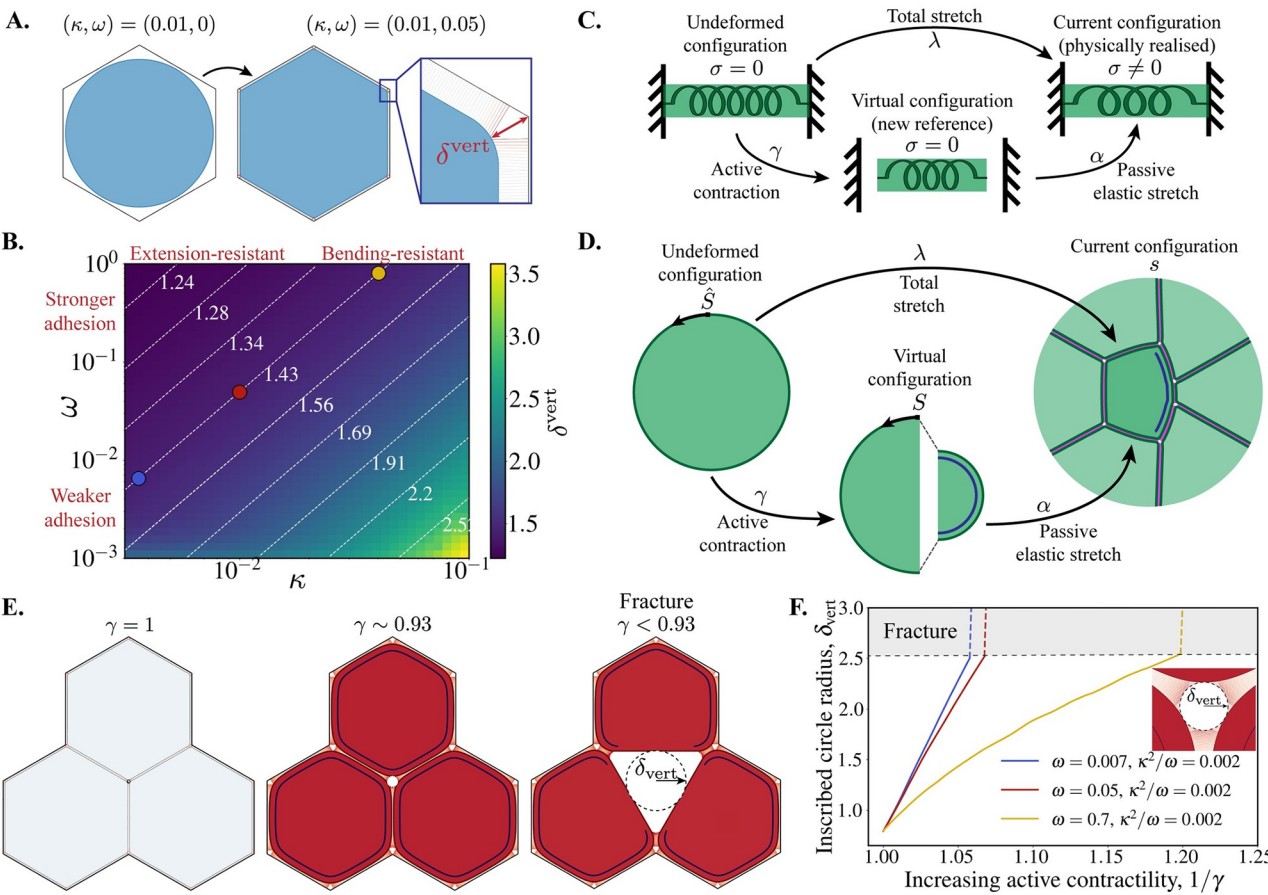

**Fig 2. Relating cell geometry to the passive and active mechanical parameters in the ACAM. (A)** A circular cell cortex is initialised with no adhesions ($\omega = 0$). The adhesion strength is quasi-statically increased to pull the cell towards the hexagonal boundary and close the vertices. **(B)** Map of parameter space showing the opening at a vertex, $\delta^{\text{vert}}$, (see **A**) relative to the parameters $\kappa$ and $\omega$ on a log scale. For each $(\kappa, \omega)$, the cell is recurrently relaxed to equilibrium with their rest lengths updated to the current length at every step, repeated until the cortex length change is less than $1 \times 10^{-4}$. Dashed white lines show isolines of fixed $\delta^{\text{vert}}$, corresponding to constant $\kappa^2/\omega$. Coloured circles show the parameter samples along the isoline used in **F**. **(C)** The three configurations of a morphoelastic spring. An active contraction would take the spring from its undeformed configuration to a new theoretical reference (virtual) configuration, where the spring is stress free. An elastic stretch, imposed by the boundary conditions, brings the spring to its current (physically realised) configuration, with non-zero stress ($\sigma \neq 0$). **(D)** A cell cortex in the morphoelastic framework: the undeformed configuration (parameterised by $\hat{S}$) is taken to the virtual/reference configuration (parameterised by $S$) by an active pre-stress e.g. from Myosin contraction ($\gamma < 1$ locally, marked by dark blue line). Forces from boundary conditions and external body loads then bring the cortex to the physically realised configuration, called the current configuration (parameterised by $s$), with an elastic stretch, $\alpha$. The total stretch, $\lambda$, at a location on the cortex is the product of active contraction and external pulling: $\lambda = \gamma\alpha$. **(E)** Snapshots from a 3-cell simulation with $(\kappa, \omega) = (0.01, 0.05)$ and increasing active cortical contractility in all cells ($\gamma$ decreasing from 1; see also S1 Movie). The tissue fractures when $\gamma < 0.93$. The dashed line shows a circle inscribed within the vertex, with radius $\delta_{\text{vert}}$. **(F)** Radius of the vertex-inscribed circle vs. the inverse magnitude of active contractility for three parameter samples on the $\kappa^2/\omega = 0.002$ isoline in passive parameter space, where $\delta_{\text{vert}}$ is constant (coloured dots in **B**).

lengths. Note that the active contraction does not produce an elastic stress if the cortex is free to contract. External physical forces and constraints may prevent the cortex from reaching its new contracted reference configuration. These external stresses and strains bring the cortex to its final, physically realised shape (parameterised by $s$) with a stretch:

$$\alpha = \frac{\partial s}{\partial S}, \qquad (9)$$

from which elastic stresses arise. The total stretch, $\lambda$, in a portion of cortex is then the product of deformation from the active contraction and external force: $\lambda = \gamma\alpha$. This framework can

therefore explicitly separate the contribution of resistance to deformation from active contractility and the material properties of the cortex (stiffness).

## Active contractility affects vertex stability and can drive material flow between junctions

The geometry of cell vertices, set by $\kappa^2/\omega$ (Fig 2B), is perturbed when cells are mechanically active. Simulating isotropic whole-cortex contractility (setting $\gamma < 1$) in a minimal 3-cell tissue, we find that active stresses perturb vertex stability. For sufficiently strong active contractility, the vertex opening is forced beyond the maximum adhesion binding length, $\delta_{max}$, and the tissue fractures (Fig 2E and S1 Movie). Tracking the size of the vertex opening as active contractility is increased, we see that larger values of the adhesion strength, $\omega$, produce more stable vertices that can sustain more active stress before fracturing (Fig 2F). This behaviour is in agreement with cell culture studies indicating that vertices are potential sites of weakness, fracturing above a critical whole-cortex contractility threshold [54]. We have provided the direct relationship between fracture behaviour and the mechanical properties of adhesions and the cortex.

If the forces acting at a vertex are not isotropic, the actomyosin cortex itself can flow past vertices (which do not act as physical barriers) and thus between junctions in the cell (S2 Movie and S1(B) Fig). If the cortex of a neighbouring cell is not flowing concomitantly, this leads to slippage between the apposed cortices in bicellular junctions. Alternatively, taking the viewpoint of a given location on a cell, the vertex can slide along the cortex past this location. This behaviour changes the length of bicellular junctions. Adhesions disconnect along the junction that is receding, while new adhesion bonds form with the cortex of the cell that is advancing. A pair of neighbouring bicellular junctions can thus commensurately increase and decrease their lengths, without compression and extension, by passing material between each other, across the vertex (S1(C) Fig). Importantly, this behaviour cannot be manifested in JVMs, where junctions behave as spring–dashpot elements connected at vertices (Fig 1C). There, cortical material is locked between vertices and junction shrinkage can happen only via contraction and viscous dissipation. In the ACAM, the cell cortex behaves as an extensible, viscous rope that is anchored by cell–cell adhesions. When adhesion bonds turnover or break, the cortices slip relative to one another. Now, junction shrinkage can be achieved by both contraction and junction slippage (which results in the passing of cortical material between junctions). The friction associated with this junction slippage depends on the adhesion timescale and is studied in further detail below.

## Implementing active neighbour exchanges in the Apposed-Cortex Adhesion Model

Active neighbour exchanges are often driven by the enrichment of subcellular components on a subset of bicellular junctions [4]. In particular, contractile Myosin II motors are commonly thought to drive junction shrinkage. We simulate this process by localising active stress on a single bicellular junction in a minimally sized tissue (Fig 3). The choice of fourteen cells allows all cells connected to the shrinking junction to be surrounded by other cells, whilst keeping the computational cost to a minimum. In order to mimic the overall tissue tension revealed by laser ablations [55, 56], we impose a low level of pre-stress, setting $\gamma = \gamma_0 = 1 - \epsilon_0$ for $\epsilon_0 \ll 1$, in all cell cortices.

Recent experimental evidence suggests that cells in the *Drosophila* germ-band enrich cortical Myosin II using a neighbour 'identity'-sensing mechanism [17, 57–59]. In particular, cells have been found to locally recruit Myosin in response to the genetic identity of their neighbours, specified by the asymmetric localisation of cell surface receptors between the apposed

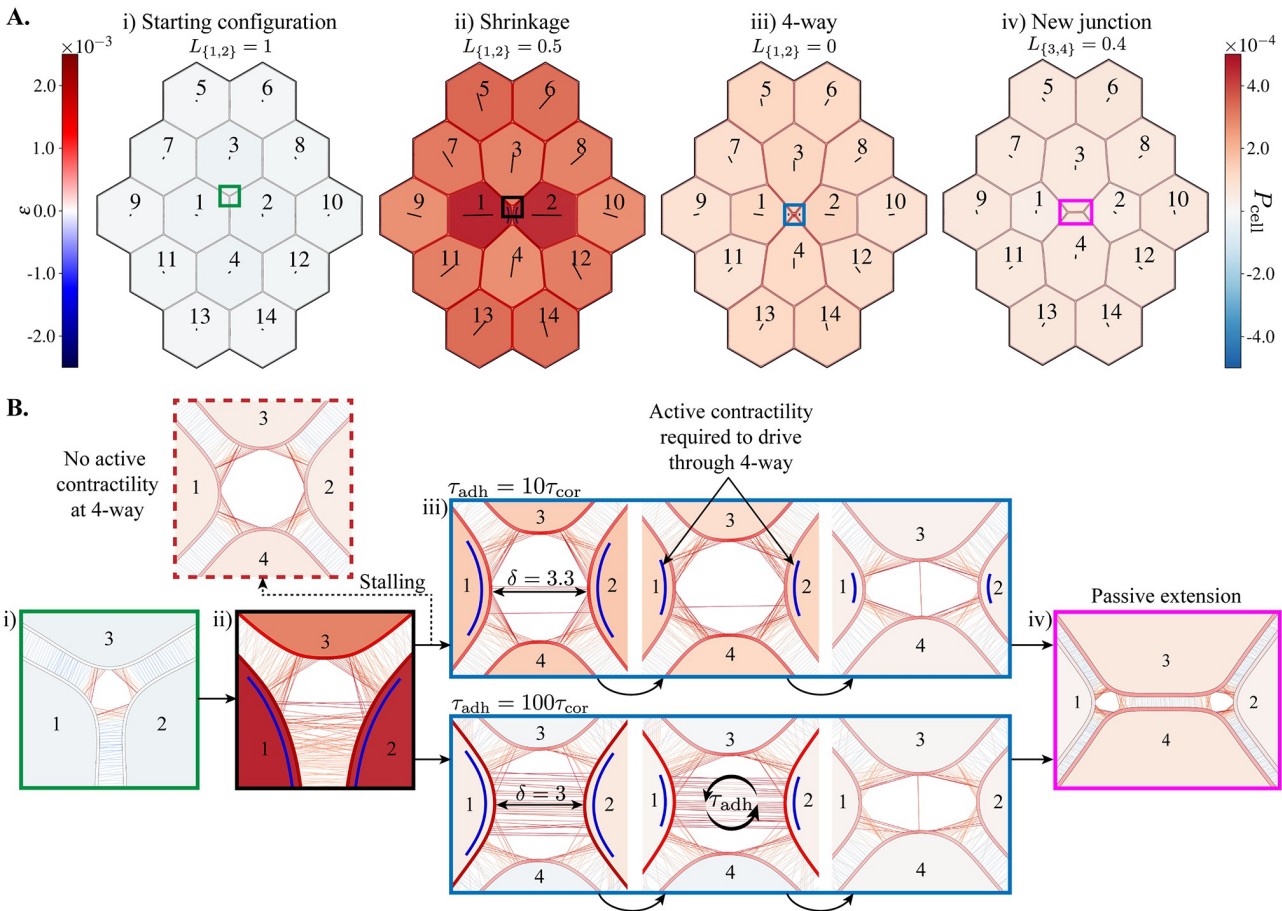

**Fig 3. Simulating an active neighbour exchange. (A)** Image sequence from S3 Movie, with $\tau_{adh} = 10\tau_{cor}$, $(\kappa, \omega) = (0.01, 0.05)$ and active contractility ($\gamma = 0.94$) in only the $\{1, 2\}$ junction. All other cortex locations (in all cells) have a low-level pre-stress, $\gamma_0 = 1 - 2 \times 10^{-4}$. Cell shading represents the magnitude of isotropic cell stress, $P_{cell}$ (see Eq (17) of S1 Text), and cortex shading represents the strain, $\varepsilon = \alpha - 1$. Lines within cells show the principal axis of cell stress. **(B)** Close-ups of cell vertices from coloured boxes in **A**. (i) Before active contractility is applied. (ii) During junction shrinkage. The dashed arrow shows the case where the active contractility is removed before the 4-way vertex is resolved and the exchange stalls. (iii) Resolution of the 4-way vertex for two choices of the adhesion timescale. (iv) Following resolution, the new junction extends passively. Cortex locations with active contractility ($\gamma < 1$) have been marked by blue lines next to the cortex.

cortices in a bicellular junction [59]. We mimic this mechanism by making the local level of active cortical contractility dependent on the identity of the cells with which these surface receptors are shared. We can then simulate the active contraction of the bicellular junction between cells 1 and 2 by setting $\gamma < 1$ in the portions of their cortices where they are within the maximum binding range of the surface receptors. For simplicity, we take this range equal to $\delta_{max}$. Biologically, cells 1 and and 2 would be of different genetic identities, sensed by their surface receptors. In the germ-band, there are stripes of identity patterns leading to cables of contractility [17]. However, for simplicity in this minimal simulation, we assume that all other cells in the tissue are passive.

### Active contractility can drive complete junction removal, while a new junction extends passively

Fig 3 shows an example simulation of an actively driven neighbour exchange (see also S3 Movie). We find that active contractility in a single junction is sufficient to shrink the junction

to zero length. The dynamics of resolving the 4-way vertex are then determined by the magnitude of active contractility and the turnover of adhesion bonds. Active contractility is required to bring the cells coming into contact (cells 3 and 4 in Fig 3Biii) close enough for an adhesion bond to form, while the magnitude of active contractility and adhesion turnover determine the rate at which this happens (discussed in further detail below).

The 4-way configuration is resolved by the appearance of a new junction, which extends passively. Then, since there are no longer adhesions between cells 1 and 2, there is no active contractility. The passive extension is due to the decreasing free energy of the system, up to the point when the bicellular junctions neighbouring the new junction form internal angles of $2\pi/3$. The extension is facilitated by, but does not require, the low-level pre-stress in all cells. Moreover, the neighbour exchange and extension relax stress at the cell level, which tends to orient towards the contracting junction (see cell shading and orientation of axes in Fig 3 and S3 Movie).

Fig 3B shows the organisation of adhesion bonds through the course of the neighbour exchange, for two values of $\tau_{adh}$. We observe more disorganised adhesion configurations as the lifetime of adhesions increases. Larger values of $\tau_{adh}$ also keep the cortices in the shrinking junction more strongly coupled, reducing the inter-cortical distance ($\delta = 3$ vs. $\delta = 3.3$, Fig 3Biii). If the active contractility is not sufficiently large compared to overall tissue tension, the inter-cortical distance can remain too large to allow adhesions to create the new junction (if $\delta > \delta_{max}$, see S12 Movie). Furthermore, cells 3 and 4 cannot come into contact until the existing adhesions between cells 1 and 2 unbind, at rate $1/\tau_{adh}$. Despite this delay, the shrinkage, resolution and extension phases are successful for all $\tau_{adh} < \infty$, as long as active contractility endures until the 4-way vertex is resolved. If the active contractility is removed before the 4-way vertex is resolved then the rearrangement stalls and the cells become stuck in the 4-way configuration (tested for 1000 simulation steps; see stalling in Fig 3B). We will next show how the adhesion timescale has important consequences for the dynamics of the process.

## Adhesion turnover modulates cell area loss and apposed-cortex slippage

Repeating the junction shrinkage simulation for several demonstrative adhesion timescale choices, we find that fast adhesion turnover ($\tau_{adh} < \tau_{cor}$) leads to significant area loss in the cells sharing the neighbouring junction (Fig 4A; compare Figs 3Aii to 4C and S3 to S4 Movies). This area loss is commensurate with increased slippage between apposed cortices in the bicellular junctions connected to the shrinking junction, indicating that the cortices of cells 1 and 2 are poorly coupled to their neighbours (Fig 4B). By contrast, when adhesion turnover is slow, adhesion bonds maintain their couplings and sliding between coupled cell cortices gradually increases the angles between adhesion bonds and the cortex (Fig 4D). This, in turn, causes adhesion forces to oppose slippage, behaving as an emergent viscous friction of coefficient $\mu_{adh} = \omega\tau_{adh}$, restricting cell–cell slippage and, therefore, the flow of cortical actomyosin relative to the neighbouring junction. Both the area loss and apposed-cortex slippage are dramatically reduced for any $\tau_{adh} \geq \tau_{cor}$, with area losses of $\sim 25\%$. It is known that intercalating cells in epithelia can exhibit area fluctuations, which have been shown to be associated with medial Myosin contractility [21]. However, when averaging cell area specifically when cells are losing a junction during convergent–extension in the *Drosophila* germ-band, as a function of the length of the shrinking junction, we find that there is also a trend of area reduction to which the model prediction is strikingly similar in magnitude (Fig 4A).

## Adhesion turnover regulates the flow of cortical material

Slippage between apposed cortices in a bicellular junction is indicative of faster flow of the liquid-like actin cortex in one of the cells sharing the junction. We quantify this by tracking the

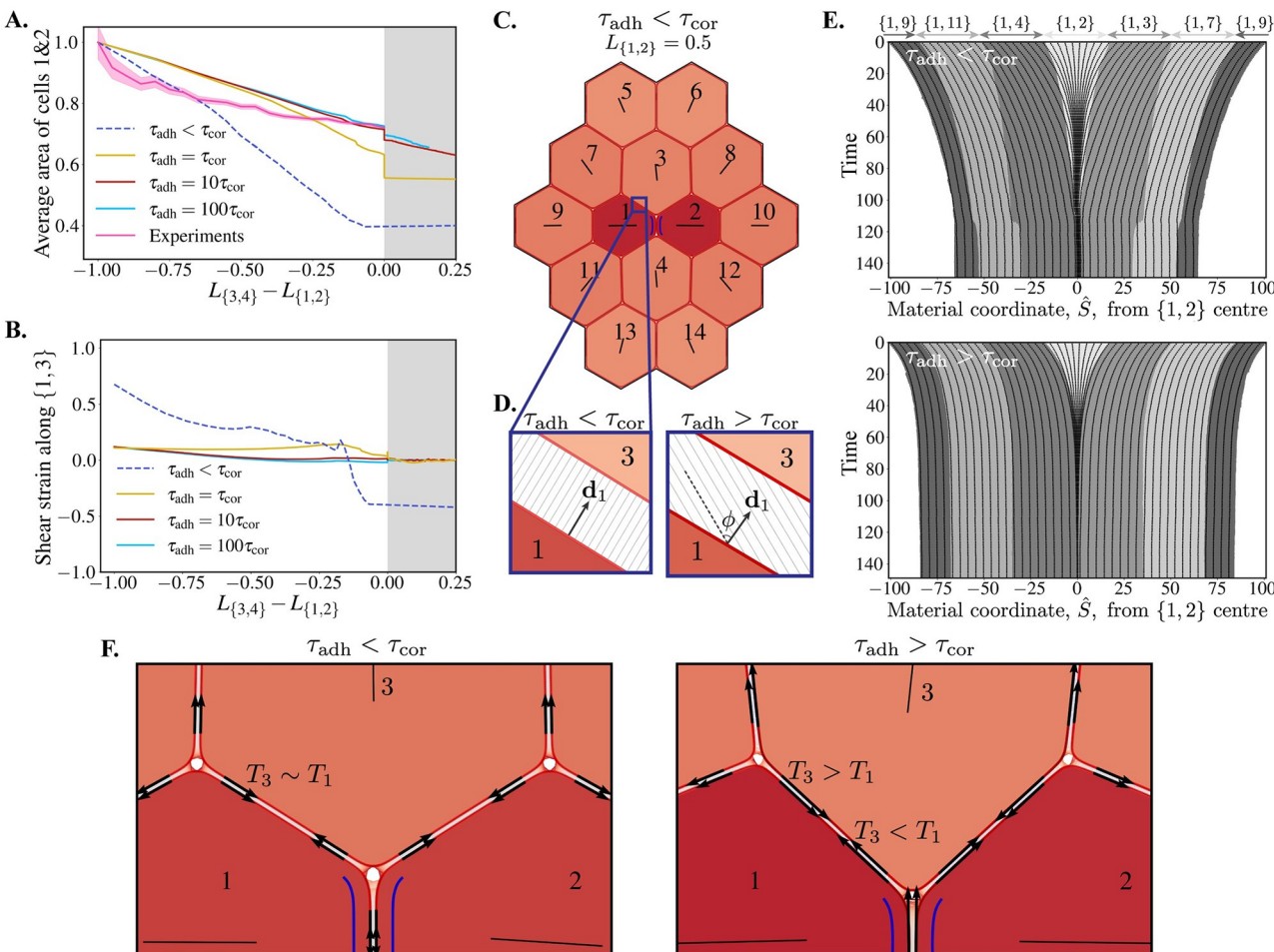

**Fig 4. The adhesion timescale regulates area loss, cell–cell slippage, material flow and tension transmission. (A)** Average area of cells 1 & 2 vs. signed length of the shrinking/growing T1 junction ($L_{\{3,4\}} - L_{\{1,2\}}$) in simulations and experiments, normalised to their initial values. Experimental data are from wild-type cells in the *Drosophila* germ-band, reanalysed from the dataset originally published in [17]. Shading represents 95% confidence intervals. **(B)** Tangential shear strain between cortices 1 and 3 at the centre of the $\{1, 3\}$ junction vs. signed length of T1 junction (as in **A**). **(C)** Snapshot from junction shrinkage simulation with fast adhesions, $\tau_{adh} < \tau_{cor}$ (see also S4 Movie), where the $\{1, 2\}$ junction has had a 50% reduction in length, comparable to Fig 3Aii. **(D)** Example snapshots of adhesions along the $\{1, 3\}$ junction in fast and slow adhesion turnover conditions. On slow timescales there is a large shear angle, $\phi$, relative to the cortex normal, $\mathbf{d}_1$. **(E)** Kymographs along the entire cortex of cell 1 during junction shrinkage, for fast (top; S5 Movie) and slow (bottom; S6 Movie) adhesion timescales. Grey shading represents each bicellular junction, with the $\{1, 2\}$ junction centred at 0 and eventually disappearing. The *x*-axis shows the reference material coordinate, *S*. Fixed material points (black lines; purple dots on cell 1 in Movies 3B and C) are tracked over the course of the simulation. Black lines crossing the grey shading boundaries indicates cortical material passing a vertex and moving between junctions. **(F)** Zoom of tissues in **C** (left; see Movies 3D and 3B) and Fig 3Aii (right; see Movies 3E and 3C) showing the different magnitude of tension, *T*, (black arrows) in the apposed cortices of bicellular junctions.

locations of material points (Lagrangian tracers) in the cell cortices. Fig 4E shows kymographs of sampled material points in the whole cortex of cell 1 over the course of a neighbour exchange. For both fast and slow adhesion turnover, we see that material points in the shrinking $\{1, 2\}$ junction collapse as the junction contracts (see also blue circles on the cortex of cell 1 in Movies 3B and C). However, this effect is exacerbated when the adhesion timescale is short relative to the timescale of the cortex, as there is minimal resistance to slippage and material is dragged from the rest of the cortex into the contracting $\{1, 2\}$ junction. This is an extreme case where there is almost no friction associated with slippage and significant cortical material flows freely between junctions. In contrast, slower adhesion turnover increases the viscous

friction coefficient between apposed cortices and prevents material transfer (notice that the solid black lines, representing material points, mostly stay within the same shaded grey region, representing junctions, in Fig 4E, lower, whereas they cross grey boundaries in Fig 4E, upper). This resembles a JVM, where there is no exchange of material between neighbouring junctions. However, rather than vertices acting as physical barriers, it is adhesion molecules that are keeping neighbouring portions of cortex coupled to prevent, but not abolish, flow.

## Adhesion turnover regulates tension transmission in an active tissue

Slower adhesion turnover prevents cortical material from being passed across vertices, reducing slippage between neighbouring cells, and provides a stronger mechanical coupling between apposed cell cortices. We explore this further by assessing the tension in apposed cortices along bicellular junctions in different adhesion timescale regimes. When the adhesion timescale is short, relative to the cortical timescale, the active stress generated in the shrinking junction is held by the elasticity in the cortices of cells 1 and 2. Fig 4F (left; see also S7 Movie) shows that cortical tension is relatively constant around the cortex of cell 1 and is of the same order as in the cortex of cell 3. In this case, the pull from the contracting junction has been transmitted around the entire cortex of cell 1. Over time, this drags material from the rest of the cortex into the shrinking junction, reducing the cell perimeter and, therefore, the area. The stress in the neighbouring cell cortices then comes from the expansion required to fill the space thus left vacant. Conversely, when adhesion turnover is slower than cortical turnover, tension is transmitted through the lingering adhesions from cell 1 to its neighbouring cortices (e.g. cell 3 in Fig 4F, right; see also S8 Movie). Here, the tension in the cortex of cell 3, for example, arises from the adhesions pulling its cortical material towards the contracting junction. Note also that, as a result of these different mechanical balances, the angles formed by junctions at the vertices differ depending on $\tau_{adh}$ (Fig 4D).

## Adhesion turnover specifies a friction that regulates tissue dynamics by resisting cell–cell slippage

By resisting slippage between apposed cortices, slow adhesion turnover is specifying a viscous friction, $\mu_{adh} = \omega\tau_{adh}$, which emerges from the dynamics of adhesion molecules. We see how this regulates the dynamics of shrinkage, resolution and extension by tracking the length of the shrinking {1, 2} junction across simulation time (Fig 5A). When adhesion dynamics are fast

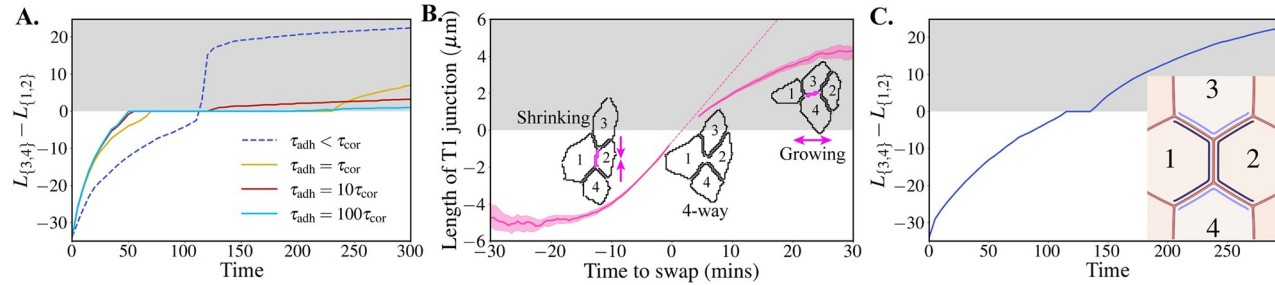

**Fig 5. The adhesion timescale regulates junction shrinkage, resolution and extension dynamics. (A)** Signed length of T1 junction (negative when {1, 2} shrinking; positive when {3, 4} growing, grey shading) vs. simulation time-step. **(B)** Experimental data of signed length of T1 junction vs. time to neighbour exchange swap for wild-type cells in the *Drosophila* germ-band. The data was extracted from experiments originally published in [25]. Lengths below 0.75 $\mu$m are not plotted, due to insufficient imaging resolution. Dashed line shows continuation of shrinkage speed from 5 mins before swap, with gradient 0.38 $\mu$m/min, which meets re-emergence of solid line after swap, indicating a minimal delay during the resolution phase. Shading represents 95% confidence intervals. **(C)** Signed length of T1 junction vs. simulation step for a simulation where contractility is applied asymmetrically along bicellular junctions neighbouring the shrinking junction. Cortices next to darker (or lighter, resp.) blue lines in snapshot have $\gamma = 0.99$ (or 0.9975, resp.). See also S9 Movie.

relative to cortex relaxation ($\tau_{adh} < \tau_{cor}$) active shrinkage is slow. This is because slippage resistance is low. The active tension is not being balanced by the one in cortices of neighbouring cells (Fig 4E), and material from the rest of the cortex in cells 1 & 2 flows into the contracting junction. However, the 4-way configuration is resolved almost immediately and followed by very fast extension, since it involves little perimeter change and the uncoupling resistance from $\mu_{adh}$ is small. Shrinkage speed increases proportionally as adhesion turnover is slowed, since lingering adhesions prevent slippage and material in-flow while effectively transmitting the active stress to the neighbouring cells. These results are robust with respect to changes in the maximum adhesion length parameter, $\delta_{max}$, as long as adhesions can stretch to accommodate the active contractility (tissue fractures for $\delta_{max} \leq 2$; see S2(A) Fig). If the active contractility is sufficiently strong to fracture the junction, the fracture hole can be repaired and the cells stall at a 4-way vertex (see S12 Movie).

Cells become jammed in the 4-way configuration for a length of time that is specified by the adhesion timescale and the magnitude of surrounding active stress. The adhesion timescale defines when cells 1 and 2 uncouple (unless the surrounding active stress is large enough to break the adhesions). Once uncoupled, the magnitude of active contractility defines how quickly the 4-way is resolved.

Our experimental data show that intercalating cells in the *Drosophila* germ-band remove a junction in roughly 20 minutes, on average, with a shrinkage rate that gradually increases to 0.38 $\mu$m/min (Fig 5B). We can match the behaviour and timescale of shrinkage in simulations by gradually increasing the strength of active contractility, mimicking gradual loading and accumulation of Myosin II motors over time (S2(B) Fig). Cells in experiments then resolve the 4-way configuration with almost no delay, which is remarkable (Fig 5B). We also find that junction extension is slower than shrinkage (Fig 5B). However, junction extension is proportionally faster, relative to shrinkage, than it is in our simulations. It is likely that there are multiple mechanisms facilitating extension, including anisotropic forces from within the bulk of the cells [10, 60]. Nevertheless, we ask whether the ACAM can reproduce the experimental shrinkage and growth rates without adding these additional forces. Indeed, we show that this can be done using a purely junctional mechanism by considering cables with asymmetric contractility within bicellular junctions. We set the cortices of cells 1, 2, 3 and 4 to be contractile when they share adhesions, creating two cables: {{1, 3}, {1, 2}, {1, 4}} and {{2, 3}, {1, 2}, {2, 4}} (Fig 5C). This represents, for example, actomyosin cables that are observed in the *Drosophila* germ-band. Following experimental predictions [17], we impose that the magnitude of contractility is larger in the cortices of cells 1 and 2. Thus the bicellular junctions {1, 3}, {1, 4}, {2, 3} and {2, 4} have asymmetric contractility—a unique capability of the apposed cortex model. In this configuration, we successfully reduce the delay in resolution and match shrinkage speed with extension (Fig 5C and S9 Movie).

## Slow adhesion turnover can promote rosette formation in an active tissue

The dynamic behaviour of adhesions becomes increasingly important in a highly active tissue, where multiple rearrangements may happen in succession. We explore this case by initialising contractility in the neighbouring {1, 3} junction after the 4-way configuration has been resolved (starting from Fig 3Aiii), but before much extension of the nascent {3, 4} junction (Fig 6). Note that this eliminates the contribution of a delay in resolving the 4-way vertex. When the adhesion timescale is of the order of the cortical timescale, the newly contracting junction facilitates extension of the nascent {3, 4} junction and there is a second successful neighbour exchange (S10 Movie). However, we can increase the penalty to junction slippage relative to contraction by slowing the adhesion turnover, increasing $\mu_{adh}$ (S11 Movie). In this

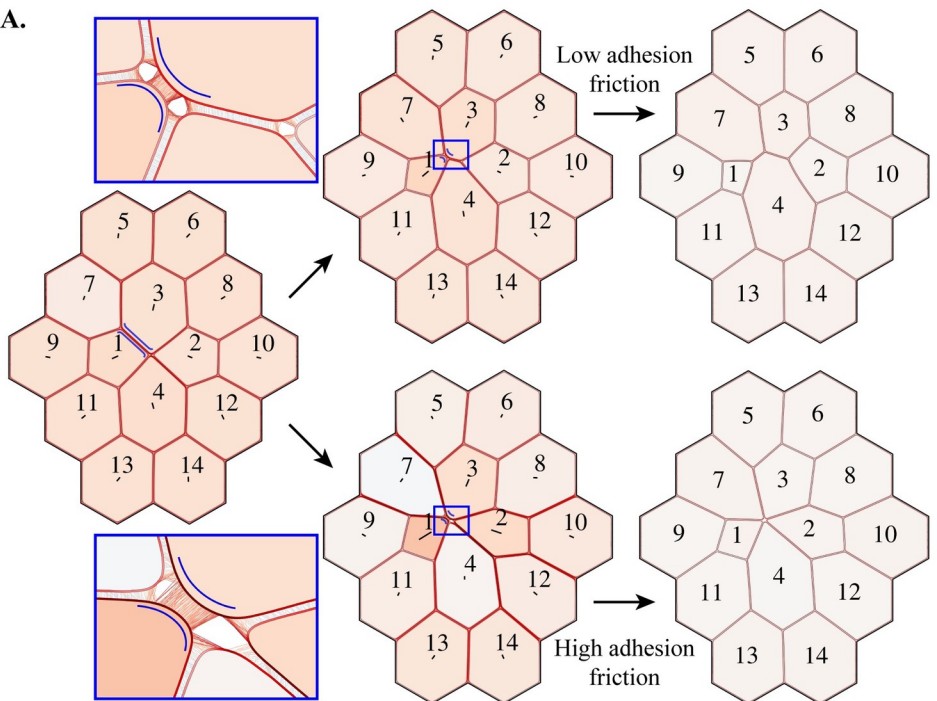

**Fig 6. Rosette formation is regulated by the resistance to cell-cell slippage. (A)** The simulation from Fig 3 was stopped just as the {3, 4} junction had formed. Active contractility ($\gamma = 0.94$) was then applied to the {1, 3} junction. If the cortical remodelling time is comparable to the adhesion turnover time ($\tau_{\text{adh}} \sim \tau_{\text{cor}}$, top; S10 Movie), the {3, 4} junction extends. However, if the adhesion timescale is much larger ($\tau_{\text{adh}} > \tau_{\text{cor}}$), preventing slippage between neighbouring cells, the {3, 4} does not resolve (bottom; S11 Movie). Instead, two vertices are merged to form a rosette structure. The rosette is stable while all cells have a small pre-stress, $\gamma_0$.

case, extension of the new junction is slower than shrinkage of the newly contracting {1, 3} junction and its two vertices are dragged together and merged. This leads to the formation of a 5-way vertex, also known as a rosette [24, 61]. This demonstrates that the adhesion timescale can promote rosette formation, not only by intuitively delaying the resolution of 4-way vertices but, by increasing the penalty to cell–cell slippage. Interestingly, the rosette appears to be indefinitely stable in the numerical simulations (running for 1000 time-steps) while there is a small pre-stretch/stress, $\gamma_0$, in all cells. Similarly to 4-way vertices, it is expected that a rosette will resolve by the growth of a new junction between two cells if these are close enough to start adhering. Since we have shown that tissue tension promotes vertex opening, it also increases the stability of this metastable state. Removing the background tension in all cells, by setting $\gamma_0 = 1$ in all cells, leads to eventual resolution of the rosette. This indicates that rosettes may be intrinsically unstable due to local edge tensions [62], but that metastability can arise if the low-level tension in the cortices of the surrounding cells prevents adhesion formation.

## Discussion

Representing the actomyosin cortex of each cell in a tissue as a continuous viscoelastic rope-loop, we have presented a novel model of active epithelial dynamics. We term this new class of model an Apposed-Cortex Adhesion Model (ACAM). Going beyond phenomenological descriptions of mechanics in vertex-based models, the ACAM can relate fundamental cell properties (e.g. the size and stability of vertices) and behaviours (e.g. neighbour exchanges and

rosette formation) to the known and measurable properties of the junctional actin cytoskeleton ($\kappa$, $\tau_{cor}$), active Myosin II contractility ($\gamma$) and adhesions ($\omega$, $\tau_{adh}$). S3 Fig outlines the causal relationships that we have identified, with a comparison to cell vertex models. Since the ACAM parameters have direct correspondence to the properties of subcelluar constituents, there is more effective control in accurately replicating known physiological conditions. The model could therefore provide mechanistic insight to other important cellular behaviours for morphogenesis, such as cell rounding and cytokinetic furrow formation during division.

The model is able to explore the physical properties of cell vertices. We predict that cell vertices are sites of mechanical vulnerability. This is in agreement with cell culture experiments, which exhibit vertex fracturing in the presence of strong Myosin II whole-cortex contractility [54], as well as experiments with *Drosophila* border cells which show that migration occurs preferentially thorough vertices [63]. Recent experimental evidence in *Drosophila* also indicates that cell vertices are mechanically unique regions in a tissue, with the localisation of vertex-specific molecules, such as Sidekick and Canoe, that both facilitate polarized cell intercalation and are important for tissue mechanics [25, 64–66]. We hypothesise that such molecules could regulate material flow past vertices, thus altering tissue dynamics as predicted here. While very difficult to encode in discrete vertex-based models, the impact of vertex-specific adhesions on the biophysical properties of a tissue will be straightforward to explore in the ACAM.

We show the importance of considering whether the actomyosin cortex can flow past a vertex, from one junction to the next, and that this can be regulated by the resistance to slippage between apposed cell cortices. Unexpectedly, this mechanism also gives rise to resistance to cell area change, which is a known but so far poorly explained property of epithelial tissues [67]. In the ACAM, these phenomena are a consequence of—and are modulated by—adhesion turnover, whose timescale, $\tau_{adh}$, defines a viscous friction, $\mu_{adh} = \omega\tau_{adh}$, that regulates the degree of slippage between apposed cortices. This term is reminiscent of protein binding drag [33] and cell–surface sliding friction [34]. Here, we characterise regimes of behaviour specified by cell–cell slippage resistance in a multicellular tissue.

In models where junctions are represented as spring-dashpot elements (JVMs; Fig 1C) [31, 41], vertices are physical barriers that block material passing between junctions, effectively imposing infinite sliding friction between apposed cortices, $\mu_{adh} \to \infty$. In more traditional Cell Vertex Models [28, 29], vertices are free to move along the cortex and penalties arise only when this results in a change of perimeter or area, effectively setting $\mu_{adh} \to 0$. The ACAM demonstrates how cell behaviours in these extremal regimes differ and provides the ability to explore intermediary regimes. Furthermore, while these models assume that tissue dynamics are governed by only cortical relaxation and/or substrate friction (see S3 Fig), there is often a lack of evidence that these are the dominant or exclusive drag terms. This work flags the little-considered friction from adhesion turnover as an important additional regulator of tension transmission and junction strain rates. We therefore propose that experimental quantification of the flow of cortical material (and, ideally, measurements of $\tau_{adh}$) could be used as a novel probe of tissue mechanics.

Many existing models of epithelial mechanics treat cell intercalations as discrete, instantaneous events and thus cannot study the complete dynamics of their behaviour. We use the ACAM to demonstrate that cell neighbour exchanges (shrinkage, 4-way resolution and extension) can be driven with only junctional contractility. This is supported by experimental observations of junctional Myosin II in shrinking junctions [14], while providing new theoretical evidence that additional mechanisms are not required. Other contributions to polarized cell intercalation observed *in vivo* include basal protrusions, first identified in vertebrate models [32, 68] and more recently in *Drosophila* germ-band extension [69]. Also in the *Drosophila*

germ-band, recent studies have indicated that medial pools of Myosin II, associated with the apical cell surface and distinct from the junctional pools, contribute to junction shrinkage [11]. Point forces at vertices and (possibly anisotropic) medial forces could be added to the ACAM to explore how these tissue-specific mechanisms act to facilitate junction shrinkage.

An additional prediction from the model is that extension of a new junction follows passively once the 4-way configuration is resolved. This passive extension can be facilitated by a small global pre-stress. *In vivo*, additional mechanisms may facilitate the elongation process and regulate its dynamics. Indeed, in the *Drosophila* germ-band, medial pools of Myosin contract asymmetrically around 4-way vertices and are thought to aid resolution and extension [10, 14, 70, 71]. Supporting this notion, we show that the relative rate of junction extension can be increased by asymmetric contractility in bicellular junctions (Fig 5C). Future work should look to match the dynamics of resolution and growth to particular experimental systems to characterise the mechanical impact of supplementary mechanisms.

Rosette structures, where four or more cells share a vertex, are found in many developing tissues [20, 23, 24, 68, 72, 73]. While these structures are thought to impact the mechanical properties of tissues [74], it is often not understood why they form with higher prevalence in particular conditions [25]. A common hypothesis is that rosettes are simply the result of sequential neighbour exchanges occurring before 4-way vertices are resolved, such that rosette prominence is a consequence of delays in neighbour exchanges. However, we provide the new prediction that rosettes form under conditions of high friction, when the penalty for cortex–cortex slippage is larger than the penalty for contraction (S11 Movie). Otherwise, in low friction regimes, the later neighbour exchange helps to resolve the first 4-way vertex and promotes growth of the new junction (S10 Movie). Simulations in large-scale models incorporating this behaviour could be used to better understand how subcellular mechanical parameters (e.g. $\mu_{\text{adh}}$) and tissue-scale geometric and topological properties (e.g. frequency of rosettes) separately contribute to tissue-level mechanical properties (e.g. shear and bulk moduli).

## Materials and methods

### STED imaging (Fig 1B)

To label the actomyosin cytoskeleton, we used the *Drosophila* transgenic strain *w, sqhTI-eGFP [29B]*, where the endogenous Myosin II Regulatory Light Chain is tagged with GFP [75].

Stage 8 *Drosophila* embryos from this strain were fixed and immunostained using standard procedures. The fixation step is 7 minutes in 40% formaldehyde. Primary antibodies were mouse anti-phosphotyrosine (Cell Signalling Jan-15, 1:1000) and rabbit anti-GFP (ab6556–25, abcam 1:500). Secondary antibodies were goat anti-mouse-Star red (Abberior, 2–0002-011-2) and goat anti-rabbit Star 580 (Abberior 2–0012-0050-8) (Life Technologies, 1:100).

Embryos were mounted on slides in Vectashield (Vector Labs) under a coverslip suspended by a one-layer thick coverslip bridge on either side. This flattened the embryos sufficiently so that all cells were roughly in the same z-plane. Prior to placing the coverslip, embryos were rolled so that their ventral surfaces were facing upwards towards the coverslip.

Embryos were imaged on an Abberior Expert Line STED microscope, in lateral depletion mode. Excitation was centred at 585 nm and 635 nm for Star 580 and Star Red respectively while 775 nm depletion was used for both colors. theAAn Olympus UPlanSApo 100x/1.40 Oil immersion lens was used on an Olympus inverted frame for all STED imaging. Xyz stacks were taken and a single plane is shown in Fig 1B at the position of adherens junctions.

### Data from time-lapse movies

Fig 1B shows segmented contours of cells at the level of adherens junctions, from time-lapse confocal movies of stage 7–8 *Drosophila* embryos, where intercalating cells are segmented and tracked through time (dataset originally published in [17]).

Biological data in Fig 4A is taken from the segmentation and tracking of *Drosophila* germ-band cells in 6 wild-type movies analysed in [17]. For each T1 event in the data (N = 5580), identified when the internal interface of a quartet of neighbouring cells changes from one pair to the other (equivalent to from cell pairs 1,2 to 3,4 in Fig 3A), the length of the shortening interface and the average area of the two cells sharing the shortening interface are extracted. Mean average area is plotted against junction length, with 95% confidence intervals shown.

Biological data in Fig 5B is extracted from the segmentation and tracking of *Drosophila* germ-band cells in 5 wild-type movies, presented in [25] (and quantified there in S9 Fig). Mean junction lengths (±95% confidence intervals) are shown for all T1 events identified between 0 and 30 minutes of germ-band extension (N = 1445).

## Supporting information

**S1 Movie. (Associated with Fig 2C).** A simulation of a tissue with three cells, where active contractility is increasing in the whole cortex of all cells ($\gamma$ decreasing from 1) until the tissue fractures. Cell shading represents the magnitude of isotropic cell stress, $P_{cell}$, and cortex shading represents the strain, $\varepsilon = \alpha - 1$.
(MP4)

**S2 Movie. (Associated with S1(B) Fig).** A simulation of a tissue with three cells, where junction shrinkage is driven by asymmetric active contractility. There is active contractility ($\gamma = 1 - 0.005 = 0.995$) in only cells 1 and 2. Cell shading represents the magnitude of isotropic cell stress, $P_{cell}$, and cortex shading represents the strain, $\varepsilon = \alpha - 1$.
(MP4)

**S3 Movie. (Associated with Fig 3).** A simulation where junction shrinkage is driven by active contractility in a single bicellular junction. There is active contractility ($\gamma = 1 - 0.04 = 0.96$) in the junction shared between cells 1 and 2 (contractility is prescribed where the two cortices are within adhesive range $\delta = 4$) and $\tau_{adh} = 10\tau_{cor}$. Cell shading represents the magnitude of isotropic cell stress, $P_{cell}$, and cortex shading represents the strain, $\varepsilon = \alpha - 1$.
(MP4)

**S4 Movie. (Associated with Fig 4C).** A junction shrinkage simulation with identical active contractility conditions to S3 Movie, applied to the {1, 2} junction. A short adhesion timescale was used, $\tau_{adh} < \tau_{cor}$, such that adhesion bonds do not persist for multiple timesteps and instead exert a mean-field force (see Eq (22) in S1 Text). Cell shading represents the magnitude of isotropic cell stress, $P_{cell}$, and cortex shading represents the strain, $\varepsilon = \alpha - 1$.
(MP4)

**S5 Movie. (Associated with Fig 4E, top).** An alternative visualisation of the simulation in S4 Movie, where $\tau_{adh} < \tau_{cor}$. In this movie, a sample of fixed material points are tracked on cortices of cells 1,2,3 and 4. We see that material in cortices 1 and 2 flows into the shrinking junction, indicating slippage behaviour between neighbouring cells.
(MP4)

**S6 Movie. (Associated with Fig 4E, bottom).** An alternative visualisation of the simulation in S3 Movie, where $\tau_{adh} = 10\tau_{cor}$. In this movie, a sample of fixed material points are tracked on

cortices of cells 1,2,3 and 4. We see that cortical material in cells 1 and 2 does not tend to flow between junction, across vertices, compared to S5 Movie.
(MP4)

**S7 Movie. (Associated with Fig 4F, left).** A magnification, around the {1, 2} junction, of the simulation shown in S4 Movie, where $\tau_{adh} < \tau_{cor}$. Black arrows show the magnitude of tension in cell junctions. We see that tension roughly equal across all junctions as the tension from the contracting {1, 2} junction is held by the whole cortices of cells 1 and 2.
(MP4)

**S8 Movie. (Associated with Fig 4F, right).** A zoom, around the {1, 2} junction, of the simulation shown in S3 Movie, where $\tau_{adh} = 10\tau_{cor}$. Black arrows show the magnitude of tension in cell junctions. We see that tension from the contracting {1, 2} junction is transmitted through adhesion bonds into the cortices of neighbouring cells.
(MP4)

**S9 Movie. (Associated with Fig 5C).** A simulation where two contractile cables are formed in the tissue: {{1, 3}, {1, 2}, {1, 4}} and {{2, 3}, {1, 2}, {2, 4}} (see dark blue lines next to cortices). However, the cables are asymmetric along bicellular junctions, with the active contractility being stronger in the cortices of cells 1 and 2 relative to cells 3 and 4; cortices next to darker (or lighter, resp.) blue lines have $\gamma = 0.99$ (or $0.9975$, resp.). Cell shading represents the magnitude of isotropic cell stress, $P_{cell}$, and cortex shading represents the strain, $\varepsilon = \alpha - 1$.
(MP4)

**S10 Movie. (Associated with Fig 6).** Multiple neighbour exchanges in a tissue where the adhesion timescale is comparable to the cortical relaxation timescale, $\tau_{adh} = \tau_{cor}$. Following removal of the {1, 2} junction and resolution of the 4-way vertex (as in S3 Movie), active contractility is engaged on the {1, 3} bicellular junction. This leads to a second neighbour exchange, with formation of a new junction between cells 4 and 7. Cell shading represents the magnitude of isotropic cell stress, $P_{cell}$, and cortex shading represents the strain, $\varepsilon = \alpha - 1$.
(MP4)

**S11 Movie. (Associated with Fig 6).** Rosette formation in a tissue with high friction, $\tau_{adh} = 100\tau_{cor}$. Following removal of the {1, 2} bicellular junction and resolution of the 4-way vertex (as in S3 Movie), active contractility is engaged on the {1, 3} bicellular junction. This leads to the formation of a rosette structure, where a vertex is shared between 5 cells. Cell shading represents the magnitude of isotropic cell stress, $P_{cell}$, and cortex shading represents the strain, $\varepsilon = \alpha - 1$.
(MP4)

**S12 Movie. (Associated with S2(A) Fig).** A simulation where junction shrinkage is driven by active contractility in a single bicellular junction, with greatly reduced maximum adhesion binding length, $\delta_{max}$. All parameters are identical to S3 Movie, except $\delta_{max} = 2$ (but $\delta_\gamma = 4$, as in S3 Movie).
(MP4)

**S13 Movie. (Associated with S2(B) Fig).** Simulation where the active junction shrinkage rate is matched experiments ($\sim 20$ mins; see Fig 5B). All parameters are identical to S3 Movie, except the magnitude of active contractility, $1 - \gamma$, in the {1, 2} junction is linearly increased until the 4-way vertex.
(MP4)

**S1 Fig. Active mechanics affect cell vertex stability and slippage between apposed-cortices.** **(A)** Representative 3-cell tissues across parameter space. The right-most example is the $\kappa^2/\omega = 0.002$ isoline with coloured circles in Fig 2B. **(B)** Snapshots from Movie 2B, driving junction shrinkage with asymmetric contractility. The whole cortices of cells 1 and 2 are contractile (dark blue line in cells shows where active contractility has been applied). Coloured dots represent fixed material (Lagrangian) points that flow past one another, between apposed cortices, demonstrating cell–cell slippage. Boundary vertices have been pinned with extra-stiff adhesions ($50\omega$; purple lines highlighted in green box) to maintain vertices at boundary angular points. Cell shading represents the magnitude of isotropic cell stress $P_{\text{cell}}$. **(C)** Kymograph along junctions $\{1, 2\}$ and $\{1, 3\}$ in the cortex of cell 1, showing the motion of material points (coloured Lagrangian markers) during the simulation shown in **B** and S2 Movie. The $x$-axis origin is at the transition from the $\{1, 2\}$ to $\{1, 3\}$ junction. Grey shading represents the extent of each junction, with darker grey showing growing $\{1, 3\}$ and lighter showing the shrinking $\{1, 2\}$. Coloured lines crossing the boundary between light/dark grey shading indicate material points flowing past the vertex, from $\{1, 3\}$ to $\{1, 2\}$.
(TIF)

**S2 Fig. Robustness of T1 dynamics. (A)** Signed length of T1 junction vs. simulation time-step (as Fig 5A) for a range of $\delta_{\max}$, with $(\kappa, \omega) = (0.01, 0.05)$, $\tau_{\text{adh}} = 10\tau_{\text{cor}}$, $\gamma = 1 - 0.04$ and $\delta_\gamma = 4$. The dynamics are robust, up to $\delta_{\max} \leq 2$ where the $\{1, 2\}$ junction fully fractures at simulation step 15 See S12 Movie. **(B)** Signed length of T1 junction vs. simulation time-step where the magnitude of active contractility ($1 - \gamma$; dashed red line) increases linearly over simulation time (see also S13 Movie). Taking the cortical timescale as $\tau_{\text{adh}} = 50$s, we infer the total shrinkage time, $T \sim 20$ mins. All other parameters match **A**, with $\delta_{\max} = 4$.
(TIF)

**S3 Fig. Putative causality networks for the Cell Vertex Model (CVM) and Apposed-Cortex Adhesion Model (ACAM).** Notice that, for the ACAM, all parent nodes are linked to model parameters that regulate subcellualr properties, such that all derived cell-level behaviours are can be traced back to subcellular mechanics. For the CVM, cell-cell tension transmission and neighbour exchange behaviours cannot be regulated in the model. Furthermore, the parent nodes of the CVM relate to cell-level, rather than subcellular, properties. Red arrows highlight loops in the network.
(TIF)

**S1 Table. Table showing parameters used in simulations.** [†] For this simulation with asymmetric contractility across some bicellular junctions, cells 1 and 2 were given $\gamma = 1 - 0.01 = 0.99$ and cells 3 and 4 were set with $\gamma = 1 - 0.0025 = 0.9975$.
(PDF)

**S1 Text. Supplementary document.** Document providing mathematical derivation of the ACAM.
(PDF)

## Acknowledgments

The computations were performed using the Cactus cluster of the CIMENT infrastructure and the authors thank Philippe Beys who manages the cluster. We thank Martin O. Lenz from the Cambridge Advanced Imaging Centre for help with STED imaging. We thank Bruno Monier for the gift of the Sqh-GFP strain. Special thanks to Claire Lye, Alexander Erlich and Karin

John for their critical reading of the manuscript and members of Bénédicte Sanson's research group for their helpful discussions.

## Author Contributions

**Conceptualization:** Alexander Nestor-Bergmann, Guy B. Blanchard, Alexander G. Fletcher, Jocelyn Étienne, Bénédicte Sanson.

**Data curation:** Alexander Nestor-Bergmann, Guy B. Blanchard, Nathan Hervieux.

**Formal analysis:** Alexander Nestor-Bergmann, Guy B. Blanchard, Alexander G. Fletcher, Jocelyn Étienne.

**Funding acquisition:** Alexander Nestor-Bergmann, Jocelyn Étienne, Bénédicte Sanson.

**Investigation:** Alexander Nestor-Bergmann, Guy B. Blanchard, Nathan Hervieux, Alexander G. Fletcher, Jocelyn Étienne, Bénédicte Sanson.

**Methodology:** Alexander Nestor-Bergmann, Guy B. Blanchard, Alexander G. Fletcher, Jocelyn Étienne.

**Project administration:** Alexander Nestor-Bergmann, Jocelyn Étienne, Bénédicte Sanson.

**Resources:** Jocelyn Étienne, Bénédicte Sanson.

**Software:** Alexander Nestor-Bergmann.

**Supervision:** Guy B. Blanchard, Alexander G. Fletcher, Jocelyn Étienne, Bénédicte Sanson.

**Visualization:** Alexander Nestor-Bergmann, Guy B. Blanchard, Nathan Hervieux.

**Writing – original draft:** Alexander Nestor-Bergmann, Guy B. Blanchard, Nathan Hervieux, Alexander G. Fletcher, Jocelyn Étienne, Bénédicte Sanson.

**Writing – review & editing:** Alexander Nestor-Bergmann, Guy B. Blanchard, Alexander G. Fletcher, Jocelyn Étienne, Bénédicte Sanson.

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
