## [Decision Letter · Decision Letter 0]

19 Oct 2021

Dear Dr Nestor-Bergmann,

Thank you very much for submitting your manuscript "Adhesion-regulated junction slippage controls cell intercalation dynamics in an Apposed-Cortex Adhesion Model" for consideration at PLOS Computational Biology.

As with all papers reviewed by the journal, your manuscript was reviewed by members of the editorial board and by several independent reviewers. In light of the reviews (below this email), we would like to invite the resubmission of a significantly-revised version that takes into account the reviewers' comments.

In addition to the reviewers' reports, I would like to add these points that I would like the authors to address:

1) I could not see in eq.2 that the adhesion lowers the overall energy of the cell membrane.

2) Why don't they consider a force-dependent dissociation rate of the adhesion molecules, ie that they dissociate when stretched ?

This is also the basis of the effective friction that such binding-stretching-unbinding dynamics induce between cells (and at cell-substrate).

See for example in the context of a particle-based monolayer model:

Garcia, Simon, et al. "Physics of active jamming during collective cellular motion in a monolayer." Proceedings of the National Academy of Sciences 112.50 (2015): 15314-15319.‏

3) This work seems relevant to discuss, in the context of how this model could be applied to biological problems:

Dai, Wei, et al. "Tissue topography steers migrating Drosophila border cells." Science 370.6519 (2020): 987-990.‏

We cannot make any decision about publication until we have seen the revised manuscript and your response to the reviewers' comments. Your revised manuscript is also likely to be sent to reviewers for further evaluation.

Sincerely,

Nir Gov

Associate Editor

PLOS Computational Biology

Mark Alber

Deputy Editor

PLOS Computational Biology

In addition to the reviewers' reports, I would like to add these points that I would like the authors to address:

1) I could not see in eq.2 that the adhesion lowers the overall energy of the cell membrane.

2) Why don't they consider a force-dependent dissociation rate of the adhesion molecules, ie that they dissociate when stretched ?

This is also the basis of the effective friction that such binding-stretching-unbinding dynamics induce between cells (and at cell-substrate).

See for example in the context of a particle-based monolayer model:

Garcia, Simon, et al. "Physics of active jamming during collective cellular motion in a monolayer." Proceedings of the National Academy of Sciences 112.50 (2015): 15314-15319.‏

3) This work seems relevant to discuss, in the context of how this model could be applied to biological problems:

Dai, Wei, et al. "Tissue topography steers migrating Drosophila border cells." Science 370.6519 (2020): 987-990.‏

Reviewer's Responses to Questions

**Comments to the Authors:**

Reviewer #1: Summary

In this manuscript, Nestor-Bermann et al develop and present the Apposed-Cortex Adhesion Model (ACAM), a new cell-based biophysical model for tissues along with several results related to cell-cell adhesion and rearrangements, of particular importance in ensuring tissues are able to undergo remodelling during many developmental processes. Many existing biophysical models for tissues, in particular vertex models, model the interface between two cells as a single edge with adhesions connecting the two cells together not explicitly modelled. This means that cell rearrangements in these models do not consider the dynamic uncoupling between two interfaces, and instead a rearrangement is often triggered by hand through a geometric argument, such as when an edge becomes small.

In the Apposed-Cortex Adhesion Model, the apical surface of each individual cell is modelled as a viscoelastic loop with both a bending and stretch stiffness, that can actively contract. When two cell interfaces are near, adhesion bonds bind the two interfaces together. These bonds are modelled by springs, attempting to keep interfaces connected, but also undergo turnover, enabling larger scale remodelling of junctions and even uncoupling between two connected interfaces as in rearrangements. Additionally, the junction is described as a continuum, enabling more complex cell shapes than in vertex models and variation along individual junctions. Model parameters are tuned to experimental data, for example, adhesion binding and unbinding rates, and actomyosin cortex turnover. Additional experiments are performed to measure spacing between two apposed cell cortices to estimate adhesive bond lengths.

Using this model, they first study the steady state geometry of a simple 3-cell tissue, predicting the spacing between cells at vertices, or 3-way junctions, as a function of adhesive stiffness and contractility. When contractility is too high, the vertex fractures and cells lose adhesion between much of their interfaces. Next, they model the process of actively driven rearrangements by increasing tension between two apposed interfaces. The tension shrinks the apposed cell interfaces which then lose adhesion and then extend passively. During this process, the cells lose some area depending on the rate of adhesion turnover, which is also observed experimentally. Additionally, rosettes, or 4+ cell interfaces may be formed when an additional rearrangement is triggered before the first is fully resolved and the adhesion turnover time is high.

Overall, this paper develops a novel biophysical model for cells and tissues that includes a more detailed description of cell-cell interfaces by including separated cortices with adhesion molecules connecting them. I feel that this model, and expansion upon it, could help our understanding of the mechanical behaviour of cell-cell junctions and rearrangements, in particular when we are interested in finer details such as the mechanics of adhesions or heterogeneity along junctions. Subject to addressing the points below, I think this paper would be worthy of publication.

Major Points

1. The results for vertex spacing and neighbour exchange are all reliant on breaking of adhesions between two cells, which depends on the parameter delta_max. Most other parameters are tuned to experiments but this seems to have been chosen arbitrarily. How do the key results of this paper depend on this parameter?

2. What defines the rest configuration of a cell? With no contraction, bending energy wants to straighten edges so expands the lengths, and this expansion is relaxed. Should cells then get infinitely bigger to reduce bending energy while relaxing stretching energy?

3. Similarly, what determines the static size of cells? If the only energy is from bending at the corners, can cells scale their lengths without changing the shape of corners and also remain stable?

4. The time scales between simulations in Fig 5B and 5C seem off. Each simulation timestep is 50s, so the total T1 would take about 300 minutes, compared to 30 minutes in experiments. Why are these differences so large?

Minor Points

1. There have been other models of cells as individual entities connected by springlike adhesions that would be worth comparison eg Nematbakhsh et al (2017)

2. I find the details of the fluid model of the cortex a bit unclear. At the start of each simulation step, the rest lengths are relaxed such that the stretch alpha = 0?

3. How do adhesions decide which node to connect to when rebinding? Is it to all nearby nodes or only the closest?

4. I think the explanation for W saturating in Eq.2 (so that these bonds are effectively broken during force balance steps) should be mentioned in the main text for clarity.

5. In Movie 2A: the cells 1 and 2 remain contractile even after they lose adhesion to each other, which is needed to complete the T1. I thought they are contractile in places where they share adhesions, so why does this happen?

6. In Movie 3A: the T1 occurs very fast, from one frame to the next cells 1 and 2 completely lose adhesion and are moved large distances. Is this what is actually simulated? This seems like a big jump, though it may be due to completely relaxing forces each time step.

7. In the T1 simulations, it appears that the high fast a_d limit gives you vertex model like behaviour, especially the “isogonal” transformations seen in Noll et al (2017), while slow a_d limit gives JVM like behaviour since there is no material flow and each edge acts elastically. This would be worth discussing since the ACAM can tune between these two models.

8. Why do the low tau_d limit material points on different apposed cells slide at different rates when their tensions are equal eg in Fig 4F?

9. Can slippage of points between cells be observed in the experimental T1 movies eg by tracking punctae of Myosin II?

10. Why are the patterns of tension required to resolve T1s quickly as in Figure 5C? Are they resolved quickly when cell tension is increased as in vertex models?

11. Can 4 way vertices be stabilised with the use of tension?

12. Why does tension stabilise rosettes in the high adhesion friction limit? Can the authors give an intuitive answer?

13. Supplementary material line 52: “update the current length of the cortex segments to the current length”. Should this be “rest length to the current length”?

Reviewer #2: The authors propose a novel mechanistic model for epithelial dynamics – specifically cell intercalation – where the key feature of the new model is the representation of cell-cell junctions not as single edges, but explicitly considering the apposed cell cortices of the two participating cells facing each other at that edge, where the individual junctional actomyosin cortex of each cell is modeled as a continuum viscoelastic rope-loop.

Let me start with the positives: I think this model is highly original and potentially valuable to the field. There are only very few published papers that consider any distinct contributions of the apposed cortices of the two participating cells, so that this Apposed-Cortex-Adhesion Model has the potential to make significant contributions to our understanding of intercalation; I think that there are a number of processes in intercalation that could involve some type of ‘heterotypic’ interaction between cortices. In terms of the theoretical modeling, the basic model is well crafted and well documented. Also, I wanted to mention that I thought the supplementary movies of the simulations were great, and I really enjoyed having data presented in this way.

However, there are a number of issues that dampened my enthusiasm. The central sticking point was that in its current form, the manuscript does not yet seem to make a very compelling case that this new (and more fine-grained) model actually provides novel insights into cellular mechanisms, specifically into the mechanisms of intercalation in the drosophila germ band. In some cases, the model merely reproduced existing knowledge (and doesn’t discuss in much detail how their findings fit in with the published literature); in other cases, the model oversimplified the biological system to the point of potentially rendering the simulation results meaningless to the biological mechanism of interest; so I would like to focus on those aspects of the model.

One of these simplifications occurs very early the manuscript and propagates from there: The authors state that “we therefore model a vertex as the geometric point where three or more cells are coupled by adhesions, rather than a material point”. But why? There is a solid body of evidence that tricellular junctions (both in mammalian systems and in drosophila) are specialized and molecularly distinct structures that do in fact mark the material point where three or more cells meet, and which likely organize true three-way adhesions (as opposed to a combination of bicellular adhesions); the manuscript’s authors obviously know this since they themselves have published on the role of sidekick in tricellular junctions. In this reviewer's opinion, the existence of tricellular connections would completely change the rationale (or even invalidate the entire premise) of Section 2.2, i.e. the energetic ‘competition’ between maximizing adhesion contact in sharp corners vs elastic penalty due to bending. Are there compelling arguments for ignoring the contributions of tricellular connections? Are the authors making the point that tricellular junction components act like combinations of bicellular junctions for the purposes of the model? This question deserves more than a passing mention in the Discussion section, and should be discussed in much greater detail when the rationale for the model components is explained.

In addition, the fact that active contractility in vertical junctions causes contractions of the junction seems really trivial; or are authors simply stating here that their more fine-grained model reproduces the behavior of the biological model (as well as the existing simpler single-edge model)? The same applies to section 2.10 (bicellular junctions have asymmetric contractility); is this simply a reproduction of the feature already known from the single-edge model in Rauzi et al. 2008, i.e. when you introduce a feature into the model that prevents or resists re-elongation of the contracted vertical junction, this promotes successful elongation of horizontal junctions?

In section 2.6, the authors describe passive extension (“due to decreasing free energy up to an internal angle of 2*pi/3”); is this actually the case experimentally, that the extension proceeds up to an internal angle 2*pi/3? And, again, this passive elongation has already been shown to work in single-edge models, it is not surprising that it would work similarly in this model; but more importantly, Collinet et al. 2015 have already shown compelling evidence that there are likely active mechanisms for horizontal junction elongation; therefore, how does the passive elongation in the apposed-cortex model contribute to our understanding of the biological system?

In section 2.7 (Adhesion turnover modulates cell area loss), the authors seem to be ignoring a body of literature (starting with Fernandez-Gonzalez 2011) that has shown that cell areas during germ band elongation undergo systematic oscillations. Isn’t it possible that the observed area loss here is merely an alignment artifact? This would mean that the experimentally observed area reduction is not necessarily a specific property of the four-way vertex resolution itself, but part of a continuum of dynamic area changes that occurs throughout the entire vertical junction contraction and horizontal junction elongation.

Reviewer #3: Summary: In this manuscript, the authors present a new computational model for describing the behaviors of intercalating epithelial cells during processes such as convergent extension. Their model, which they call the “apposed cortex adhesion model” (ACAM), represents a considerable advance over older junction-vertex models. In older models, complex cell-cell interfaces were represented as single springs connecting nodes; and while these models could often recapitulate the phenomenological behaviors of intercalating cells, they were not very informative about the underlying causes of these behaviors. With ACAM, the authors created a two-dimensional model in which each actomyosin cortex is represented as a continuous rope-like structure with differential adhesion between neighboring cells. As their model parameters have actual subcellular processes (e.g., myosin contraction and membrane turnover) as analogs, this allows them to use this model to better predict how cells will behave in different contexts, and which subcellular processes are important for which behaviors. While the manuscript was written with Drosophila germband extension in mind, this model should be applicable to other cellular contexts as well. Importantly, the authors do make several interesting predictions/conclusions using this model. 1) They show that intercalation can be explained completely by junctional behaviors, and that new edge formation does not require an active process. 2) They are able to explain decreases in apical cell area during intercalation in terms of increased slippage between apposed cortices, which to me is a novel prediction. And 3) They are able to recapitulate rosette formation under certain conditions with high adhesion friction, suggesting that rosettes are really a result of active edge contraction winning out over passive new edge formation.

Issues:

My major issue with the paper (if it can be called a major issue) is the question of whether medial myosin forces need to be incorporated into the model. I understand this would be a major change to the model, and it is probably beyond the scope of a revision. Upon reading the Discussion section again, I do believe that they address the most obvious questions concerning how medial myosin might affect the model.

Line 43. The authors use the phrase “can cortical actomyosin flow past a cell vertex into neighboring junctions…” (or some similar phrase), several times throughout the paper. I’m not sure I understand what this phrase means. Are they asking whether the vertex is a physical boundary that stops myosin from moving past it? Or are they asking whether myosin can move along paths other than the junctions? They should rephrase here and throughout the paper to make their meaning more clear.

Line 50: I think the authors are being a bit hyperbolic when they state “However, the physical roles that these subcellular features take and whether they are strictly necessary or merely supportive, are not known.” First, the word “features” is too vague. If they are referring to myosin, as there has been an immense amount of work showing that myosin is necessary and sufficient for cell intercalation in many systems, so this statement isn’t true. I think it would help the paper if they reworded this paragraph to be more specific about what is truly unknown at this point.

Minor issues:

Line 78, 86, etc: Should “continuum” be “continuous”? It sounds a bit strange…

Figure 4E. Is the top axis label correct? Should it read: {1,9} {1,7} {1,3} {1,2} {1,4} {1,11} {1,9}

**Have the authors made all data and (if applicable) computational code underlying the findings in their manuscript fully available?**

Reviewer #1: Yes

Reviewer #2: Yes

Reviewer #3: Yes

PLOS authors have the option to publish the peer review history of their article (what does this mean?). If published, this will include your full peer review and any attached files.

Reviewer #1: No

Reviewer #2: No

Reviewer #3: No
---

## [Decision Letter · Decision Letter 1]

6 Jan 2022

Dear Dr Nestor-Bergmann,

We are pleased to inform you that your manuscript 'Adhesion-regulated junction slippage controls cell intercalation dynamics in an Apposed-Cortex Adhesion Model' has been provisionally accepted for publication in PLOS Computational Biology.

Best regards,

Nir Gov

Associate Editor

PLOS Computational Biology

Mark Alber

Deputy Editor

PLOS Computational Biology

Reviewer's Responses to Questions

**Comments to the Authors:**

Reviewer #1: I thank the authors for their response. Their revised manuscript now addresses all of the previous concerns I had and makes for a clearer and stronger paper. In particularly, the parts I found a little unclear have been more thoroughly explained, and my concern over a key model parameter, the maximum binding length, has been shown to not affect results except in extreme cases. I believe the manuscript is worthy of publication.

Reviewer #3: I believe the author’s ACAM model is an important step forward for accurately describing the molecular bases of cell- and tissue-level behaviors during epithelial reorganization. My reservations with respect to the original manuscript were relatively minor, and I believe the authors have adequately addressed my points through textual changes to the Introduction and Discussion section. My main critique (which was noted by at least one other reviewer) was that very little attention is paid to medial myosin in this model, despite a lot of experimental work on this topic. While the authors agree that incorporating medial myosin into the model is a short-term goal for future studies (perhaps focusing on other tissues), they stress that medial myosin does not appear necessary to recapitulate germband extension (which is an interesting finding in and of itself). In the end, I agree with the authors that for this initial description of the ACAM model, they should focus on only the minimal components necessary to recapitulate real-world observations. Furthermore, the new edits make the manuscript much easier to understand, particularly when it comes to describing material being moved across vertices between junctions. I feel that the authors have put in a good faith effort to address all reviewer comments, and I would recommend this manuscript for publication in PLoS Computational Biology.

**Have the authors made all data and (if applicable) computational code underlying the findings in their manuscript fully available?**

Reviewer #1: Yes

Reviewer #3: None

PLOS authors have the option to publish the peer review history of their article (what does this mean?). If published, this will include your full peer review and any attached files.

Reviewer #1: No

Reviewer #3: No

---

## [Editor Report · Acceptance letter]

24 Jan 2022

PCOMPBIOL-D-21-01600R1 

Adhesion-regulated junction slippage controls cell intercalation dynamics in an Apposed-Cortex Adhesion Model

Dear Dr Nestor-Bergmann,

I am pleased to inform you that your manuscript has been formally accepted for publication in PLOS Computational Biology. Your manuscript is now with our production department and you will be notified of the publication date in due course.

With kind regards,

Livia Horvath
